# Quantification of FRET-induced angular displacement by monitoring sensitized acceptor anisotropy using a dim fluorescent donor

Danai Laskaratou [1], Guillermo Solís Fernández[2], Quinten Coucke [2], Eduard Fron[2,3], Susana Rocha [2], Johan Hofkens[2], Jelle Hendrix [2,4] & Hideaki Mizuno [1✉]

Förster resonance energy transfer (FRET) between fluorescent proteins has become a common platform for designing genetically encoded biosensors. For live cell imaging, the acceptor-to-donor intensity ratio is most commonly used to readout FRET efficiency, which largely depends on the proximity between donor and acceptor. Here, we introduce an anisotropy-based mode of FRET detection (FADED: FRET-induced Angular Displacement Evaluation via Dim donor), which probes for relative orientation rather than proximity alteration. A key element in this technique is suppression of donor bleed-through, which allows measuring purer sensitized acceptor anisotropy. This is achieved by developing Geuda Sapphire, a low-quantum-yield FRET-competent fluorescent protein donor. As a proof of principle, $Ca^{2+}$ sensors were designed using calmodulin as a sensing domain, showing sigmoidal dose response to $Ca^{2+}$. By monitoring the anisotropy, a $Ca^{2+}$ rise in living HeLa cells is observed upon histamine challenging. We conclude that FADED provides a method for quantifying the angular displacement via FRET.

---

[1] Laboratory for Biomolecular Network Dynamics, Biochemistry, Molecular and Structural Biology Section, Department of Chemistry, KU Leuven Heverlee, Belgium. [2] Chem & Tech-Molecular Imaging and Photonics, Department of Chemistry, KU Leuven Heverlee, Belgium. [3] KU Leuven Core Facility for Advanced Spectroscopy, KU Leuven Heverlee, Belgium. [4] Dynamic Bioimaging Lab, Advanced Optical Microscopy Centre and Biomedical Research Institute, Hasselt University, Agoralaan C (BIOMED), Diepenbeek, Belgium. ✉email: hideaki.mizuno@kuleuven.be

Förster resonance energy transfer (FRET) is a process by which excited-state energy of a donor luminophore is transferred to an acceptor chromophore through dipole-dipole coupling[1]. Since the coupling efficiency correlates with the proximity and relative angular orientation of transition dipole moments, FRET serves as a tool to analyze molecular distance and conformation[2]. FRET-based biosensors have been developed to read out conformational alteration of a protein module, so called sensing domain, upon certain biological phenomenon, such as Ca$^{2+}$ signaling[3–5], cAMP signaling[6,7], small G-protein activation[8–10] and protein phosphorylation[11–13]. Genetically encoded FRET biosensors have been designed by connecting a pair of fluorescent proteins to the sensing domain as donor and acceptor (intramolecular FRET sensor), which are now widely used in life sciences for live cell functional imaging in cultured systems[14–16], as well as in vivo[17,18].

A crucial step in designing a FRET biosensor is devising optimal sensing domains that show good contrast in FRET efficiency upon conformational change. Proximity-based sensors are relatively simple, since FRET efficiency increases once the donor and acceptor come closer. On the other hand, the impact of relative orientation is not straightforward, as it is difficult to predict how it affects FRET efficiency upon conformational alternation. In addition, the change in relative orientation might counteract the change in proximity, and consequently the effects can be cancelled out mutually. An optimized linker (EeVee linker) inducing a large proximity difference has been reported[19]. Conversely, a system highlighting the impact of relative orientation is still lacking, despite its potential as an alternative mode to read out FRET. This system would be complementary to proximity-based sensors, since it would provide information on the donor-acceptor angular displacement. In fact, there is a class of biosensors where change in donor-acceptor relative orientation may outweigh distance modulation[20]. In these cases, a method designed to read out angular displacement would be highly advantageous.

In this study, we establish a method to quantify the relative orientation between donor and acceptor transition dipole moments upon intramolecular FRET by detecting sensitized acceptor anisotropy. We named this technique FADED (FRET-induced Angular Displacement Evaluation via Dim donor). FADED is insensitive to distance changes, because sensitized acceptor anisotropy only reflects the angular displacement from donor to acceptor dipole moments. Since the method relies on the quantification of sensitized acceptor anisotropy, detection of pure acceptor signal without donor bleed-though is crucial. To minimize the bleed-through, we engineered and used a dim fluorescent protein with low fluorescence quantum yield as a FRET donor. Intriguingly, in the case of intramolecular FRET, a low donor quantum yield should not compromise FRET efficiency, since the acceptor is located around the Förster distance ($R_0$); the rate constant of the energy transfer $k_T(R)$ is linked to the intrinsic rate constant of excited-state decay of the donor ($k_D$), according to the following equation[21]:

$$k_T(R) = k_D \left(\frac{R_0}{R}\right)^6 \tag{1}$$

Although the Förster distance is shorter for the donor with lower quantum yield ($Q_D$), the influence is rather weak since it is proportional to the one sixth power of quantum yield, as shown in the following equation:

$$R_0 \propto \left(\kappa^2 n^{-4} Q_D J\right)^{\frac{1}{6}} \tag{2}$$

where $\kappa^2$ is the orientation factor, $n$ is the refractive index of the medium, and $J$ is the spectral overlap integral.

Dim fluorescent proteins have already been used as FRET acceptors in fluorescence lifetime imaging to minimize the bleed-through of acceptor signal into the donor channel[22–24]. However, to the best of our knowledge, no dim fluorescent protein has been used as a FRET donor before, probably due to the counter-intuitive approach. A factor hindering realization of this idea is the lack of dim-fluorescent proteins with an appropriate absorption/emission range for FRET donor. Although there are several naturally occurring dim-fluorescent proteins (also called chromoproteins), most of their absorption bands are in yellow-red range, making them unsuitable as donors. Here we developed a dim-fluorescent donor protein by site-directed mutagenesis using Sapphire (H9-40) as a template. Sapphire shows a large energy gap between absorption (400 nm) and emission (510 nm) due to excited state proton transfer (ESPT), which is beneficial for avoiding direct excitation of an acceptor. Employing Venus as the acceptor, efficient FRET from the dim-fluorescent donor is observed. By using this system, we evaluated the concept of FADED. As a proof of principle, we designed a Ca$^{2+}$ biosensor and used it to demonstrate the applicability of FADED in live cell imaging.

## Results

**Development of a dim-fluorescent Sapphire mutant.** The chromophore of fluorescent proteins is protected from the environment by the surrounding β-barrel. Since the naked GFP chromophore exhibits extremely weak fluorescence with lifetime of sub-picosecond[25], it is clear that the β-barrel significantly suppresses non-radiative decay pathways. Residue 145 on the β-barrel is located close to the chromophore, and mutations on it have been shown to greatly affect fluorescence lifetime in eYFP[26]. Since eYFP and Sapphire are both mutants of *Aequorea victoria* GFP (aqGFP), we reasoned that mutations on residue 145 can have a similar effect on Sapphire. Consequently, we mutated residue F145 of Sapphire (Supplementary Fig. 1a) to D, G, S, T, and W, to develop a fluorescent protein with short fluorescence lifetime. For Sapphire-F145D, absorption was invisible at the band corresponding to the chromophore (Supplementary Fig. 1b), and hence it was rejected as a folding-impaired mutant. Fluorescence lifetimes of other mutants were determined by time correlated single photon counting (TCSPC) (Table ST1). Sapphire-F145W showed biexponential decay with an amplitude-weighted average lifetime of 0.92 ns (Fig. 1a), which is significantly shorter than Sapphire (3.3 ns, Fig. 1b), whereas lifetimes of the other three mutants were comparable to that of Sapphire (2.7, 2.8 and 2.9 ns for Sapphire-F145G, Sapphire-F145T and Sapphire-F145S, respectively) (Supplementary Table 1). We named Sapphire-F145W "Geuda Sapphire" (GeuSap), after the sapphire gemstone with milky-white color, and further characterized this as a dim-fluorescent Sapphire mutant.

GeuSap had a relatively broad absorption band peaked at 400 nm, with a tiny shoulder around 500 nm, which was similar to that of Sapphire (Fig. 1c). The former band corresponds to the protonated chromophore of aqGFP whereas the latter to the anionic one. The extinction coefficient of GeuSap at 400 nm was determined as $3.4 \times 10^4\,M^{-1}\,cm^{-1}$ (Supplementary Table 1), which was equivalent to that of Sapphire ($3.5 \times 10^4\,M^{-1}\,cm^{-1}$). The proper folding ratio for GeuSap was 54%, similar to that for Sapphire (57%) (Supplementary Table 2).

The emission spectrum of GeuSap was acquired upon excitation at 400 nm, after adjusting the optical density to 0.1 (Fig. 1d). GeuSap fluorescence intensity was only about one fifth of Sapphire, and the spectrum was slightly broader (Fig. 1d, inset). The emission maximum appeared at 510 nm, which corresponds to the emission band of the anionic form of Sapphire

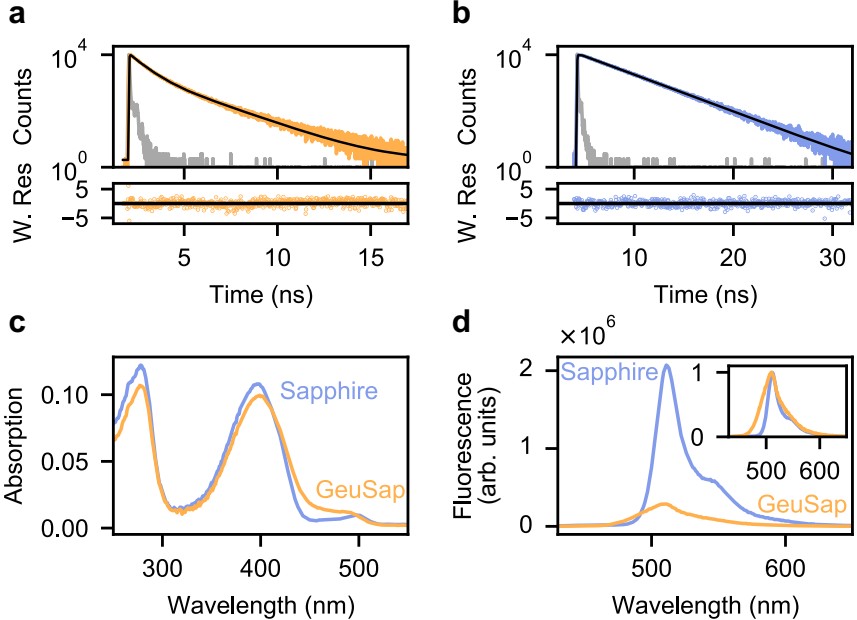

**Fig. 1 Spectral characteristics of GeuSap and Sapphire. a** Fluorescence decay of GeuSap (orange) on 400 nm excitation and 510 nm detection, measured with TCSPC. The model used for fitting (black) is a bi-exponential function with reconvolution, yielding two lifetimes $\tau_1 = 0.58$ ns (70%) and $\tau_2 = 1.7$ ns (30%). IRF is in gray. Weighted residuals are shown in the bottom panel ($\chi^2 = 1.2$). **b** Fluorescence decay of Sapphire (blue). The model used is a mono-exponential function with reconvolution (black), $\tau = 3.3$ ns and $\chi^2$ is 1.08. **c** Absorption spectra of GeuSap and Sapphire. **d** Emission spectra of GeuSap and Sapphire upon excitation at 400 nm. Optical density at the excitation wavelength was adjusted to 0.1 for both samples. Inset shows normalized emission spectra. W. Res.: weighted residuals. Source data are provided as a Source Data file.

formed by ESPT. The quantum yield of GeuSap was determined to be 0.12, using Sapphire (0.64) as a reference[27]. Rate constants for radiative decay ($k_r$) and nonradiative decay ($k_{nr}$) were calculated from the quantum yield and lifetime. The $k_r$ value of GeuSap was similar to that of Sapphire ($1.3 \times 10^8$ s$^{-1}$ and $1.9 \times 10^8$ s$^{-1}$, respectively). In contrast, the $k_{nr}$ of GeuSap was $9.7 \times 10^8$ s$^{-1}$, which was almost one order of magnitude larger than that of Sapphire ($1.1 \times 10^8$ s$^{-1}$) (Supplementary Table 2). We concluded that the shorter lifetime of GeuSap was due to its fast nonradiative decay, whereas the rates of radiative decay are comparable between GeuSap and Sapphire.

**Achieving efficient FRET with the dim-fluorescent donor.** FRET is competitive with radiative and non-radiative relaxation of donor. The FRET efficiency ($E_T$) is described by the following equation:

$$E_T = \frac{1/\tau_T}{1/\tau_T + 1/\tau_d^0} = 1 - \frac{\tau_d^A}{\tau_d^0} \qquad (3)$$

where $\tau_T$ is the time constant of FRET, $\tau_d^0$ is the amplitude-weighted average lifetime of donor in the absence of acceptor, and $\tau_d^A$ is the amplitude-weighted average lifetime of donor in the presence of the acceptor. Smaller $\tau_d^0$ gives lower $E_T$, but significant $E_T$ is expected if $\tau_T$ is equivalent to $\tau_d^0$ or shorter. Despite its short lifetime (0.92 ns), GeuSap can be a competent donor in FRET systems if the acceptor is located close enough, so that the energy transfer takes place in the order of tens-hundreds of picoseconds. To validate the concept of using the dim-fluorescent protein as a donor, we designed an efficient FRET system by concatenating GeuSap (donor) with Venus (acceptor), after truncating the C-terminus of the donor and the N-terminus of the acceptor by 11 and 5 amino acids, respectively (GeuSV11.5; Supplementary Fig. 2). A similar FRET system using Sapphire as donor was also designed as a reference (SV11.5). The absorption

spectrum of GeuSV11.5 had two maxima at 400 nm and 515 nm, corresponding to the absorption of GeuSap and Venus, respectively (Fig. 2a). The intensity ratio of the peak at 515 nm over 400 nm was 3.0. Considering that the extinction coefficient of Venus at 515 nm ($9.2 \times 10^4$ M$^{-1}$cm$^{-1}$)[28] is 2.7 times higher than that of GeuSap at 400 nm, close to 1:1 stoichiometry of the donor and acceptor was assumed for GeuSV11.5. The absorption spectrum of SV11.5 was similar to GeuSV11.5 (Fig. 2a). Emission spectra of GeuSV11.5 and SV11.5 were acquired upon 400 nm excitation after adjusting their optical density to 0.1 at 400 nm, which corresponded to a concentration of 2.8 μM. We also acquired the emission spectrum of the same concentration of Venus as a control of acceptor cross excitation, with which no significant emission signal was detected upon excitation at 400 nm (Fig. 2b, top panel). The peak intensity of GeuSap11.5 was slightly lower than that of SV11.5, but the difference was only 15%. The emission spectra of GeuSV11.5 and SV11.5, excited at 400 nm, were similar to that of Venus, excited at 490 nm (Fig. 2b, bottom panel). These observations indicated that the dim-fluorescent GeuSap worked efficiently as a FRET donor in GeuSV11.5.

Linear unmixing analysis of the GeuSV11.5 emission signal revealed that the donor peak at 510 nm was only 2.6% of the acceptor peak at 530 nm (Fig. 2c). We interpreted this result as predominant emission from the acceptor sensitized by efficient FRET, with negligible contribution from the donor. Further analyses of donor signal fractions at respective wavelength (Fig. 2d) revealed that donor bleed-through was insignificant in the main emission range of the acceptor (above 530 nm). There is a clear contrast to the results of the reference sample (SV11.5), where substantial donor contribution was seen (donor peak was 10.4% of the acceptor peak) (Fig. 2c), and a shoulder donor peak appeared in the acceptor emission range at around 550 nm (Fig. 2d).

To obtain direct evidence of FRET with GeuSV11.5, we performed an acceptor photobleaching experiment. GeuSV11.5

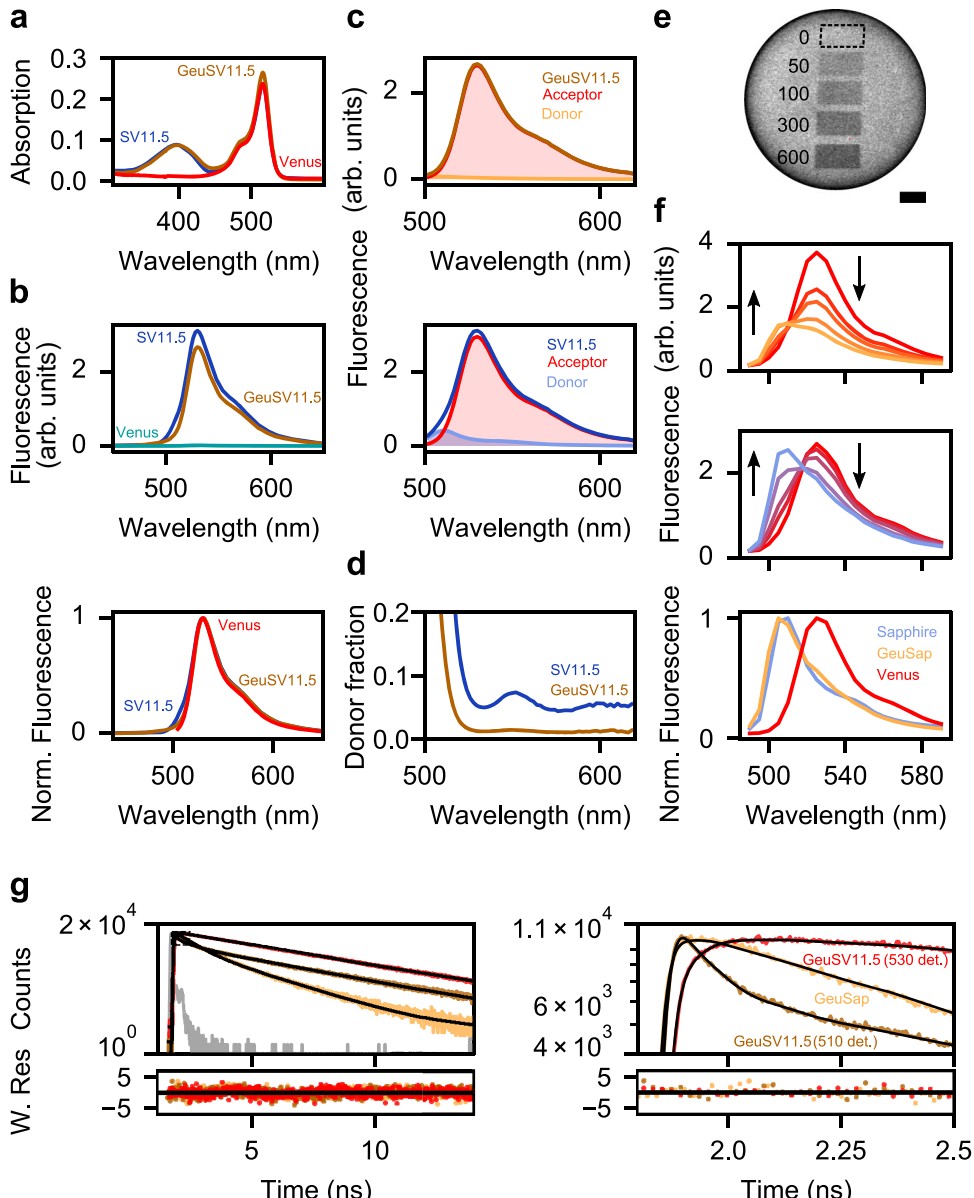

**Fig. 2 Characterization of the GeuSV11.5 FRET system. a** Absorption of GeuSV11.5 and SV11.5 adjusted to 0.1 at 400 nm, and Venus absorption adjusted accordingly to match the OD of SV11.5 at 516 nm. **b** Top: Emission spectra of GeuSV11.5, SV11.5, and Venus, corresponding to the absorption shown in **a** and excited at 400 nm. Bottom: Normalized emission spectra of GeuSV11.5 (400 nm excitation), SV11.5 (400 nm excitation), and Venus (490 nm excitation). **c** Linear unmixing performed on GeuSV11.5 (top) and SV11.5 (bottom). **d** Fractions of donor to total signal in the case of GeuSV11.5 and SV11.5, calculated based on the linear unmixing result. **e** GeuSV11.5 immobilized on a Ni-NTA bead and imaged with a laser scanning microscope under 400 nm excitation. Dark rectangles correspond to increasing photobleaching cycles performed on Venus with a 488-nm laser. The experiment was repeated 3 times with essentially the same results. Scale bar is 10 μm. **f** Emission spectra of GeuSV11.5 (top), and SV11.5 (middle) recorded under 400 nm excitation, each curve corresponding to the non- and photobleached areas shown in **e**. Arrows indicate progression of bleaching. Bottom panel shows normalized emission spectra of donors alone (GeuSap, Sapphire), and acceptor alone (Venus). **g** Fluorescence decays and fittings of GeuSap (510 nm detection, $\chi^2 =$ 1.2), GeuSV11.5 (510 nm detection, $\chi^2 = 1.2$), and GeuSV11.5 (530 nm detection, $\chi^2 = 1.15$). Excitation was at 400 nm. The dashed rectangle corresponds to the zoomed-in figure on the right. For fitted lifetimes and amplitudes, refer to Supplementary Table 3. Norm. Fluorescence: normalized fluorescence; W. Res.: weighted residuals. Source data are provided as a Source Data file.

and SV11.5 were immobilized on Ni-NTA beads, and their emission spectra were acquired with a laser scanning microscope under 400 nm laser illumination (Fig. 2e, f). Before photobleaching, both SV11.5 and GeuSV11.5 showed similar emission spectra to Venus. Photobleaching of Venus in GeuSV11.5 was performed by repeated laser scanning at 488 nm. Acceptor photobleaching with increasing bleaching cycles could be monitored as the subsiding of the peak corresponding to Venus at 530 nm in both samples (Fig. 2f). With SV11.5, recovery of donor signal at

510 nm, which corresponded to the emission maximum of Sapphire, was obvious upon acceptor photobleaching. This was also observed in the case of GeuSV11.5, although the signal after the photobleaching was weaker due to the lower quantum yield of GeuSap. In both GeuSV11.5 and SV11.5, the spectrum becomes similar to that of the donor after 600 bleaching cycles, although not identical because of incomplete acceptor photobleaching. This data indicates that FRET takes place in GeuSV11.5 upon the excitation of dim donor, GeuSap, at 400 nm.

Further, to quantify FRET, we monitored the lifetime of donor in the presence and absence of the acceptor by TCSPC (Fig. 2g). GeuSap (free donor) was excited at 400 nm and detected at 510 nm, yielding an amplitude-weighted lifetime of 0.92 ns. GeuSV11.5 (donor in the presence of acceptor) was measured under the same conditions, and yielded a markedly diminished lifetime of 0.22 ns after excluding the acceptor component (Fig. 2g and Supplementary Table 3). For comparison, we measured GeuSV11.5 at 530 nm detection as well, corresponding to the maximum emission of the acceptor. The amplitude weighted lifetime in this case was 2.4 ns, which corresponds to the acceptor. Based on the above amplitude-weighted lifetimes of donor in the presence and absence of the acceptor, we calculated the FRET efficiency of GeuSV11.5 as 76%, with a time constant of $\tau_T = 0.29$ ns (Eq. 3). Thus, we concluded that efficient FRET occurred in GeuSV11.5 upon excitation of the dim-fluorescent donor at 400 nm, resulting in strong sensitized emission from the acceptor at 530 nm.

**Steady-state anisotropy upon FRET.** FRET is expected to reduce the steady-state anisotropy, owing to the different orientation between the donor and acceptor transition dipole moments. We evaluated this change by comparing the emission anisotropy spectra of GeuSV11.5 and GeuSap (Fig. 3b). The anisotropy of GeuSap throughout the main emission band (500–580 nm) was consistently 0.35 (Fig. 3b). The anisotropy spectra of Geusap and GeuSV11.5 were found to be similar below 500 nm, where donor signal is dominant. The donor fraction became smaller at longer wavelengths. The anisotropy of GeuSV11.5 dropped steeply and reached a plateau of 0.14 at around 530 nm or longer, where the signal was exclusively from the acceptor. We interpret this big anisotropy drop as a result of FRET-induced angular displacement.

**Influence of rotational diffusion on anisotropy.** The sensitized acceptor anisotropy depends also on depolarization due to rotational diffusion of the fusion protein. To estimate the influence of molecular diffusion of GeuSV11.5, time-resolved anisotropy analysis was performed. The time-resolved anisotropy is described by the following equation:

$$r(t) = r_0 e^{-\frac{t}{\theta}} \tag{4}$$

where $r_0$ is the fundamental anisotropy and $\theta$ is the rotational correlation time. Upon direct excitation of the acceptor at 485 nm, exponential decrease of anisotropy with $\theta = 34$ ns was observed (Fig. 3d, blue line). In the case of donor excitation at 405 nm, rapid decrease of anisotropy, attributable to angular displacement via FRET[29], was observed prior to depolarization (Fig. 3d, purple line). The corresponding time constant could not be determined, since it was faster than the time resolution of our system. From the subsequent slow phase, we determined $\theta = 27$ ns. This was in good agreement with the result of direct acceptor excitation, suggesting that the rotational diffusion of GeuSV11.5 is in the order of 30 ns.

**Quantifying the angular displacement by FADED.** The steady-state anisotropy ($r$) of spherical molecules follows the Perrin equation:

$$r = r_0 \cdot \frac{1}{1 + \tau/\theta} \tag{5}$$

The fundamental anisotropy $r_0$ is an intrinsic property of the system under study. It reflects the angle between absorption and emission transition dipole moments, without taking into account any depolarization processes during the excited state (e.g.

Brownian motion). In the case of sensitized acceptor anisotropy via our FRET system, the donor-acceptor angular displacement is incorporated in the fundamental anisotropy (Fig. 3c). The fluorescence lifetime of the acceptor was determined as $\tau = 3.1$ ns from the decay of the total fluorescence intensity upon direct acceptor excitation (Fig. 3e). Based on the $\theta$ and $\tau$ values determined upon direct acceptor excitation, the steady-state anisotropy was calculated to be 92% of the fundamental anisotropy, meaning that the influence of the rotational diffusion was only 8%. From the steady-state anisotropy of GeuSV11.5 above 530 nm ($r = 0.14$), its fundamental anisotropy was calculated as $r_0 = 0.15$.

The fundamental anisotropy is described by the following equation:

$$r_0 = 0.4 \left( \frac{3\cos^2\beta - 1}{2} \right) \tag{6}$$

where $\beta$ is, in this case, the angle between the donor absorption and acceptor emission transition dipole moments. Using the fundamental anisotropy of GeuSV11.5, we find an angular displacement of $\beta = 40°$. Thus, we concluded that FADED could be used to monitor the angular displacement between the donor and acceptor.

**Engineering Ca$^{2+}$ sensors for FADED imaging.** As an application of FADED, we designed a Ca$^{2+}$ indicator using GeuSap as donor and cp173-Venus as acceptor, imitating the backbone of Yellow Cameleon YC2.60[5], and named it Geuda Cameleon GeuSapVC2.60 (Fig. 4a). As a reference, a sensor using Sapphire as donor was also designed (SapVC2.60). The emission maximum of Ca$^{2+}$-free form of GeuSapVC2.60 was at 528 nm, corresponding to the acceptor, while the donor signal appeared as a tiny shoulder at shorter wavelengths (Fig. 4b). The FRET efficiency was calculated as 40% from TCSPC measurements (Supplementary Fig. 6 and Supplementary Table 4). The donor contribution became obvious by applying spectral unmixing. The donor peak intensity at 510 nm was 0.48 times that of the acceptor (Fig. 4d, left panel), which is significantly lower than that of SapVC2.60 (1.04) (Fig. 4c, e, left panel). Upon addition of Ca$^{2+}$, significant increase in FRET efficiency was observed through increased acceptor signal, accompanied by reciprocally decreased donor signal (Fig. 4d, e, right panel). The FRET efficiency was 52% (Supplementary Fig. 6 and Supplementary Table 4). The resulting donor over acceptor intensity ratio was 0.19 and 0.40 for GeuSapV2.60 and SapVC2.60, respectively.

A major concern for acceptor anisotropy imaging is donor bleed-through into the acceptor signal. To evaluate its impact, the fraction of donor signal at respective wavelengths was calculated from the unmixing data (Fig. 4f). The donor fraction of Ca$^{2+}$-free form of GeuSapVC2.60 was consistently about 20% in the wavelength range above the emission maximum (528 nm). In the case of Ca$^{2+}$-bound form, the value went down to less than 10%. Complementary cumulative distribution analysis (Fig. 4g, Supplementary Fig. 4) revealed that the fraction of donor bleed-through above 530 nm was 20% and 9% for Ca$^{2+}$-free and Ca$^{2+}$-bound forms, respectively. Comparison with the reference sample (SapVC2.60; 29% and 14%, respectively) revealed a significant bleed-through reduction, hence validating the choice of a dim donor.

To characterize GeuSapVC2.60 as a Ca$^{2+}$ indicator, we performed a Ca$^{2+}$ titration in vitro (Fig. 5a). At low Ca$^{2+}$ concentration, the steady-state anisotropy reached its maximum value of 0.10 ($r_{free}$). We subsequently observed a marked drop upon increasing the Ca$^{2+}$ concentration in the buffer, with a plateau value of $-0.03$ ($r_{bound}$). The concentration dependency

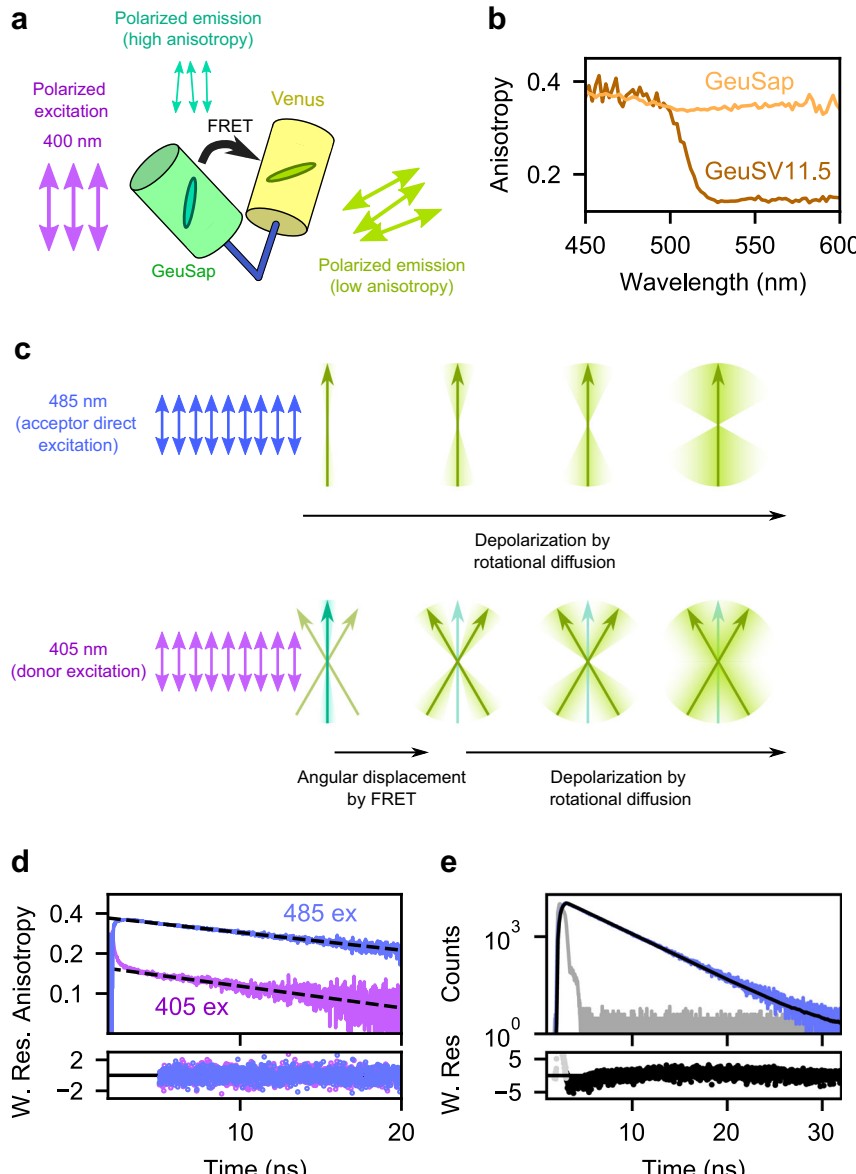

**Fig. 3 GeuSV11.5 anisotropy. a** Cartoon representation of GeuSV11.5 under polarized excitation. **b** Steady-state anisotropy spectra of GeuSap and GeuSV11.5. **c** Top: schematic drawing of direct acceptor excitation of GeuSV11.5 by linearly polarized light. The yellow-green arrow represents the emission dipole moment of the acceptor. After absorption of polarized light, the polarization of the emitted light is gradually lost as a result of rotational diffusion. Bottom: donor excitation of GeuSV11.5 by linearly polarized light. FRET is accompanied by angular displacement from donor (green arrow) to acceptor (yellow-green arrows) dipole moments. Anisotropy of the sensitized acceptor reflects initially the angular displacement, followed by gradual depolarization due to rotational diffusion. **d** Time-resolved anisotropy of GeuSV11.5 under donor (405 nm; purple) or direct acceptor (485 nm; blue) excitation. The decay after 5 ns was fitted to a mono-exponential model, and yielded $\theta = 27$ ns (sensitized acceptor) and $\theta = 34$ ns (acceptor excited directly). **e** Fluorescence decay of GeuSV11.5 under 485 nm excitation (blue). The decay was fitted to a mono-exponential model with reconvolution (black) and yielded $\tau = 3.1$ ns with $\chi^2 = 1.29$. W. Res.: weighted residuals. Source data are provided as a Source Data file.

followed a monophasic sigmoidal function with apparent dissociation constant ($K'_d$) of $71 \pm 3$ nM and Hill coefficient of $1.9 \pm 0.2$ (Fig. 5a, Supplementary Table 3). Furthermore, we performed an in vitro pH titration on GeuSapVC2.60 (Supplementary Fig. 5) and confirmed that it performs stably at pH 7.0 or higher. We also made a sensor with lower affinity for $Ca^{2+}$, named GeuSapVC3.60, by substituting glutamine for glutamate placed at the $3^{rd}$ EF-hand motif of calmodulin (E104Q). GeuSapVC3.60 also showed monophasic sigmoidal response to $Ca^{2+}$, with maximum and minimum anisotropy values of 0.10 and $-0.02$, respectively. The $K'_d$ of GeuSapVC3.60 was $502 \pm 11$ nM with Hill coefficient of $1.6 \pm 0.1$.

As mentioned above, the anisotropy in vitro could have been affected by depolarization due to rotational diffusion. To estimate this influence, we measured the time-resolved anisotropy and fluorescence decay of GeuSapVC2.60, upon direct excitation of the acceptor. In the $Ca^{2+}$-free case, we found a rotational correlation time of $\theta_{free} = 28$ ns and a fluorescence lifetime of $\tau_{free} = 3$ ns (Fig. 5b). Plugging these numbers into Perrin equation (Eq. 5), the fundamental anisotropy of the $Ca^{2+}$-free state is found to be $r_{0,free} = 0.11$, 10% more than the steady state anisotropy ($r_{free} = 0.10$). In the case of the $Ca^{2+}$-bound form of the sensor, we found $\theta_{bound} = 33$ ns and $\tau_{bound} = 3.0$ ns (Fig. 5c).

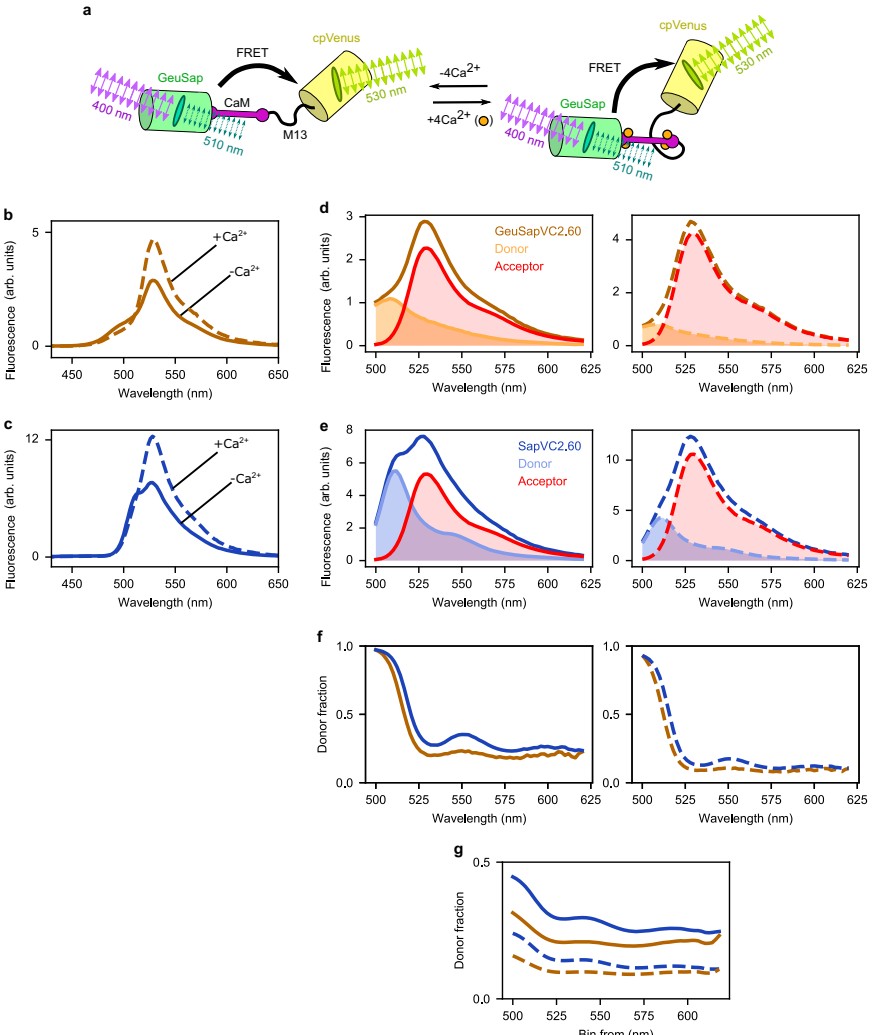

**Fig. 4 Spectral properties of GeuSapVC2.60. a** Cartoon representation of GeuSapVC2.60 in its Ca²⁺-free and -bound form under polarized illumination. **b**, **c** Emission spectra of GeuSapVC2.60 (**b**) and SapVC2.60 (**c**), under zero (solid line) and saturated (dashed line) Ca²⁺ conditions. **d**, **e** Linear unmixing of donor and acceptor contributions in emission spectra of GeuSapVC2.60 (**d**) and SapVC2.60 (**e**), Ca²⁺-free (left) and Ca²⁺-bound form (right). **f** Donor fraction (of total signal) plotted based on the linear unmixing of GeuSapVC2.60 (orange) and SapVC2.60 (blue), Ca²⁺-free (left) and Ca²⁺-bound (right) forms. **g** Fraction of donor signal over total cumulative distribution of GeuSapVC2.60 (orange) and SapVC2.60 (blue) for Ca²⁺-free (solid lines) and Ca²⁺-bound conditions (dashed lines). Source data are provided as a Source Data file.

Together with $r_{bound} = -0.03$, these yield a fundamental anisotropy of $r_{0,bound} = -0.033$ for the Ca²⁺-bound state, which is 10% less than the steady state anisotropy. When expressing the sensor in living cells, slower diffusion (namely larger $\theta$) due to higher viscosity might influence the $r$ value[30–33]. However, $r$ converges to $r_0$ even in the extreme case of $\theta = \infty$, and therefore the error never becomes more than 10%.

Although the use of GeuSap has suppressed donor bleed-through, there is still some residual contribution in the acceptor signal, which can now easily be eliminated mathematically (refer to method section for details). This compensation allowed us to precisely calculate $\beta$, the angle between donor absorption and acceptor emission dipole moments. After subtracting donor bleed-through, the acceptor anisotropy values become $r'_{free} = 0.038$ and $r'_{bound} = -0.068$ for the Ca²⁺-free and Ca²⁺-bound form of GeuSapVC2.60, respectively. Since it has been reported that the sensor domain of Ca²⁺-bound GeuSapVC2.60 adopts a rigid conformation[34], it is meaningful to calculate $\beta$ for the Ca²⁺-bound form, based on $r'_{bound}$. The fundamental anisotropy of Ca²⁺-bound GeuSapVC2.60 was calculated to be $r'_{0,bound} =$

$-0.074$ by substituting $r'_{bound}$, $\tau_{bound}$, and $\theta_{bound}$ in Eq. 5. Using $r'_{0,bound}$, we then determined $\beta_{bound} = 62.7°$ from Eq. 6.

**FADED imaging in living cells.** HeLa cells expressing Geuda Cameleon were subjected to live cell imaging for cytosolic Ca²⁺. Sensors were localized mainly in cytosolic regions (Fig. 5d left panel, 5e left panel), although weak fluorescent signal was also detected in the nucleus. Autofluorescence was insignificant compared to fluorescence originating from the sensor (Supplementary Fig. 7). We also performed in situ titration for GeuSapVC2.60 and GeuSapVC3.60 and found Hill coefficients and K'ₐ values in good agreement with in vitro titration (Supplementary Fig. 8).

We then performed an experiment to monitor the histamine-induced Ca²⁺ response of HeLa cells (Fig. 5d, e, Supplementary Movie 1, 2). For cells expressing GeuSapVC2.60, the anisotropy under resting conditions was 0.10 and went down to a plateau level of 0.02 upon histamine challenging[35] (Fig. 5d, Supplementary Movie 1). After 7 min the anisotropy gradually rose, followed by going down and up repetitively. The Ca²⁺ response was also

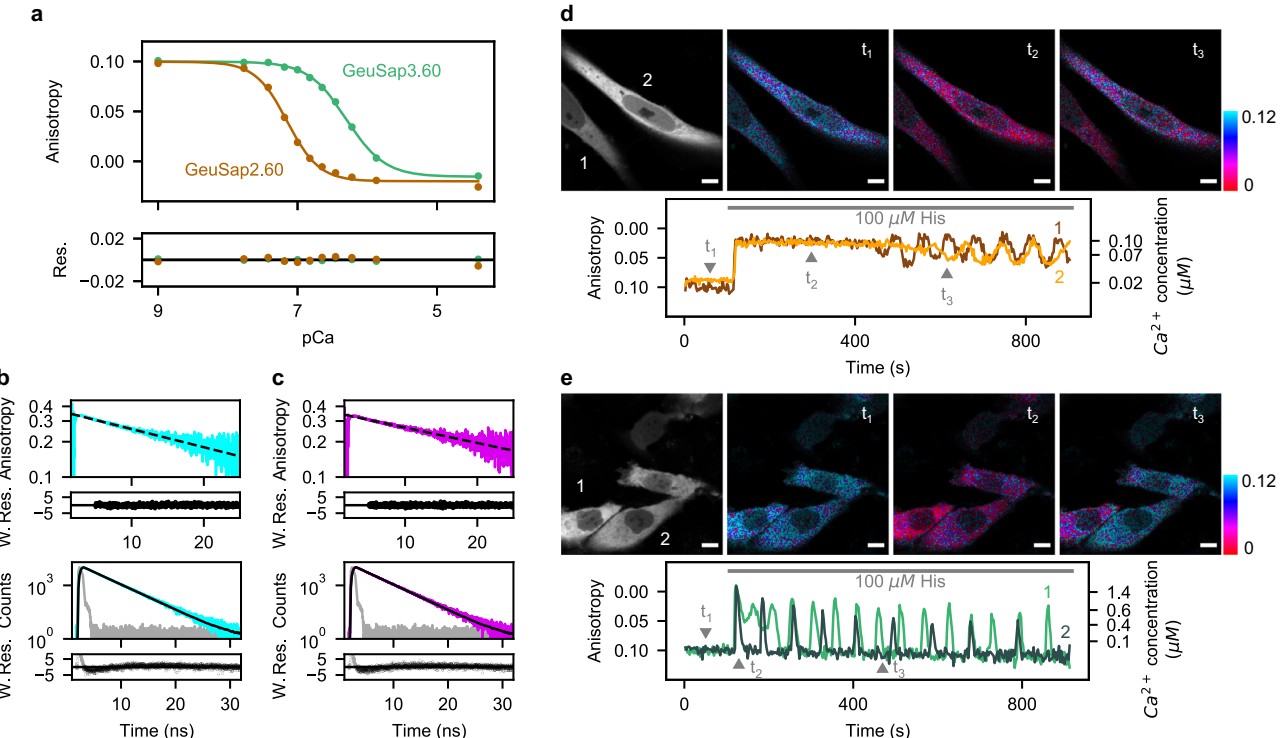

**Fig. 5 Live cell Ca$^{2+}$ imaging by monitoring acceptor anisotropy. a** Anisotropy Ca$^{2+}$ titration for GeuSapVC2.60 and GeuSapVC3.60 for calibration, fitted with Hill function. Obtained fitted values for GeuSapVC2.60 were $n = 1.9 \pm 0.2$, $K'_d = 71 \pm 3$ nM, $r_{max} = 0.10 \pm 0.00$, $r_{min} = -0.02 \pm 0.00$, and for GeuSapVC3.60 $n = 1.6 \pm 0.1$, $K'_d = 502 \pm 11$ nM, $r_{max} = 0.10 \pm 0.00$, $r_{min} = -0.02 \pm 0.00$. Residuals of the fit are shown below the main plot. **b** Top: time-resolved anisotropy decay of GeuSapVC2.60 Ca$^{2+}$-free form, excited at 485 nm. The decay was fitted to a mono-exponential model after 5 ns (dashed line), which yielded $\theta = 28$ ns. Bottom: Fluorescence lifetime decay of the same sample excited at 485 nm, fitted to a mono-exponential model (black line). The fitting yielded $\tau = 3.05$ ns and $\chi^2 = 1.25$. **c** Same plots as in b for GeuSapVC2.60 Ca$^{2+}$-bound form. In this case $\theta = 33$ ns and $\tau = 3.02$ ns with $\chi^2 = 1.15$. **d** Imaging of live HeLa cells expressing GeuSapVC2.60. Top: Stacked fluorescence image of the first 100 s (left), and anisotropy images displayed in IMD mode corresponding to timepoints indicated (right). Bottom: Time trace of fluorescence anisotropy calculated from cells indicated on the left top panel. HeLa cells were stimulated with 100 μM histamine at $t = 120$ s. Right axis indicates Ca$^{2+}$ concentrations calibrated with the titration curve shown in **a**. Signal-to-noise ratio for parallel and perpendicular channel was calculated as $17 \pm 2$ and $14 \pm 2$, respectively (Supplementary Fig. 7b and c). **e** Imaging of cells expressing GeuSapVC3.60, similar as in **d**. Scale bar represents 10 μm. For panels **d** and **e**, imaging of transfected cells was repeated at least 10 times, yielding basically the same results. Res.: residuals; W. Res.: weighted residuals. Source data are provided as a Source Data file.

observed in HeLa cells expressing GeuSapVC3.60, the lower affinity version of the sensor (Fig. 5e, Supplementary Movie 2). The expression pattern and the anisotropy value under the resting conditions were the same as that of GeuSapVC2.60. Upon histamine challenging, the periodical drop in anisotropy was observed, returning to the same basal level as before the stimulation. These results indicated that cytosolic Ca$^{2+}$ concentration in HeLa cells was tens of nanomolar under the resting conditions, and showed repetitive change between around one hundred nanomolar to a few micromolar upon histamine challenging. We, thus, concluded that cytosolic Ca$^{2+}$ concentration could be successfully quantified by FADED.

## Discussion

Due to its sensitivity, reversibility, and its potential for studying biomolecular interactions in living cells, FRET has become a well-established platform for developing genetically encoded biosensors. Although FRET is dependent on both fluorophore distance and orientation, among other factors, probes developed so far are relying on the former. In this study, we developed FADED, a system for monitoring angular displacement via intramolecular FRET by detecting sensitized acceptor anisotropy. As opposed to classical FRET biosensors, FADED has the property of reflecting changes in relative orientation between donor and acceptor,

regardless of their proximity. Consequently, this mode of FRET quantification presents information complementary to distance, which was so far unavailable. Moreover, in some cases where the change in orientation outweighs change in distance[20], we speculate that FADED-type sensors can prove beneficial over conventional distance-based biosensors, and provide meaningful biological insight.

An additional perk of FADED lies in employing fluorescence anisotropy, an absolute quantity that does not depend on measuring parameters. This allows for easy correlation between in vitro calibration and imaging in live cells. This is a striking advantage over commonly used ratiometric FRET quantification, which only offers a relative measure, varying with experimental conditions. Fluorescence lifetime imaging of FRET donor is an alternative method for absolute quantification, but it requires special instrumentation, such as pulsed lasers and a TCSPC system. In addition, longer acquisition times are typically required to accumulate enough photons, which results in lower time resolution. On the contrary, acceptor anisotropy imaging can be performed with most conventional fluorescence microscopes by adding orthogonally polarized beam splitters. For all the above reasons, FADED is a quantification method of donor-acceptor relative orientation with higher time resolution and minimum system investment. Although in this work we used pulsed lasers and a confocal setup, these modalities are certainly not required

for FADED. The technique is also applicable to confocal setup with continuous wave lasers, widefield microscopes, or even two-photon setups.

Anisotropy has been used for the detection of FRET between identical molecules (homo-FRET). In fact, homo-FRET has been successfully employed in the development of biosensors, with the potential of facilitating multiplexed FRET imaging[36,37]. However, homo-FRET does not aim to quantify the sensitized acceptor anisotropy since signals from donor and sensitized acceptor are indistinguishable. Jovin's group has introduced a concept for detecting FRET between different species (hetero-FRET) by emission anisotropy[38], but the idea of quantifying angular displacement was never explored. Methods based on emission anisotropy proposed by Jovin's group have been attempted by the following three research groups. Piston's group has reported that high-contrast imaging by anisotropy FRET was exceptional for screening purposes[39,40]. They chose Cerulean and Venus as a FRET pair and optimized the excitation wavelength to avoid cross-excitation of the acceptor, while eliminating the donor bleed-through from the sensitized acceptor signal was out of their scope. Ameer-Beg's group has employed eGFP and mRFP1 as a FRET pair for sensitized acceptor anisotropy imaging, with a Raichu-type FRET sensor[41–43]. The well-separated absorption/emission bands of eGFP and mRFP1 ensured minimal donor bleed-through, which inevitably resulted in smaller spectral overlap, and, hence, less efficient FRET. The group of Heikal developed mCerulean-mCitrine-based biosensors to study macro-molecular crowding in cells[44,45]. They used time-resolved anisotropy to analyze the rotational dynamics of the probes, but eliminating donor background and subsequent quantification of angular displacement were not in the scope of their research. Geuda Sapphire, developed in the present study, together with Venus, form a FRET system with large spectral overlap, no acceptor cross-excitation, and minimized donor bleed-through. These advantages, taken together, were incorporated into FADED to make angular displacement quantification possible.

In general, sensitized acceptor anisotropy also depends on rotational diffusion of the sensor. However, in the case of our protein-based FRET system, the influence of diffusion was marginal (less than 10%) due to the high molecular weights of fluorescent proteins. FADED requires detection of pure FRET signal by avoiding donor bleed-though. This was achieved by engineering a fluorescent protein mutant with low quantum yield, which conserved its competence as a donor. This way, the donor bleed-through could be reduced to insignificant levels in the case of the efficient FRET system (GeuSV11.5). In the case of the FRET-based $Ca^{2+}$ indicator (GeuSapVC2.60), we could observe anisotropy difference between $Ca^{2+}$-bound and $Ca^{2+}$-free form, in spite of some remaining donor bleed-through (10–20%). The difference in donor bleed-through between GeuSV11.5 and GeuSapVC2.60 is due to different FRET efficiencies (76% for GeuSV11.5, 40% for GeuSapVC2.60 $Ca^{2+}$-free form, and 52% for GeuSapVC2.60 $Ca^{2+}$-bound form). However, even in the Geu-SapVC2.60 $Ca^{2+}$-free form the donor bleed-through does not exceed 20% in the acceptor emission band (530–620 nm).

Low anisotropy of the sensitized acceptor is often explained as a result of energy transfer from a highly polarized donor to an acceptor whose orientation is unconstrained. We think this is the case for the $Ca^{2+}$-free form of GeuSapVC2.60. The acceptor anisotropy value close to zero (0.038) can be due to complete depolarization, which is consistent with the reported highly flexible structure of $Ca^{2+}$-free calmodulin in solution[46]. On the other hand, calmodulin together with the target peptide M13 adopt a compact rigid conformation upon binding of $Ca^{2+}$ [34].

Accordingly, we assumed that the orientation of the acceptor upon $Ca^{2+}$ binding is rather constrained. This assumption is consistent with the negative acceptor anisotropy of the $Ca^{2+}$-bound GeuSapVC2.60 (-0.068), which can be explained only by the angular displacement above the magic angle (54.7°), but not by depolarization due to unhindered acceptor motion. We concluded that the acceptor orientation was unconstrained in the absence of $Ca^{2+}$, whereas in the presence of $Ca^{2+}$, cylindrically symmetric distribution of the acceptor with the fixed angle of 62.7° was expected around the donor absorption dipole moment.

In conclusion, FRET-induced angular displacement could be quantified successfully by FADED, a technique which monitors sensitized acceptor anisotropy. Development of the dim-fluorescent protein, GeuSap, and using it as a donor allowed us to measure sensitized acceptor anisotropy under minimal influence of acceptor cross-excitation, donor bleed-through, and depolarization by rotational diffusion. In proteins adopting a rigid conformation, the anisotropy value can be used to estimate the relative angle between donor and acceptor. The acceptor anisotropy of flexible proteins is expected to be close to zero due to fluctuation-induced depolarization. As an application of our concept, the cytosolic $Ca^{2+}$ response could be quantified in living cells with our $Ca^{2+}$ sensor, Geuda Cameleon GeuSapVC2.60 and GeuSapVC3.60, by monitoring the sensitized acceptor anisotropy. FADED paves the way for a mode of FRET sensors that focus on relative orientation between donor and acceptor, rather than their proximity.

## Methods

**Gene construction.** Sapphire (H9-40) on pRSET was a kind gift of Prof. Roger Y. Tsien. To introduce F145D, F145G, F145T, F145S, F145W and A206K mutations, site directed mutagenesis[47] was performed.

Venus (gift from Prof. Atsushi Miyawaki) fused to GeuSap (GeuSV11.5) or Sapphire (SV11.5) were constructed on pRSETB vector as follows: first, C-terminally truncated (Δ11) GeuSap or Sapphire was amplified by PCR, with the sense primer containing a BamHI site in frame with the polyhistidine tag on the vector and the antisense containing an XhoI site. N-terminally truncated Venus (Δ5) was amplified by PCR, with the sense primer containing an XhoI site and the antisense containing a stop codon and a HindIII site. The fragments were then sequentially ligated into pRSETB vector (Invitrogen, Carlsbad, CA). This configuration of donor-acceptor concatenation was inspired from Cy11.5, the chimeric protein composed of two aqGFP mutants, eCFP and Venus, with which optimal FRET with $\tau_T$ of 66 ps has been reported[48]. GeuSap, Sapphire, eCFP and Venus are all aqGFP mutants, hence GeuSV11.5 and SV11.5 are expected to show similar spatial arrangement as Cy11.5.

For the construction of the Cameleon-based sensors for bacterial expression, YC2.60/pcDNA3 and YC3.60/pcDNA3 (a kind gift of Prof. Atsushi Miyawaki) were used as a starting point. An additional mutation, A206K, which has been shown to favor monomeric state of GFP variants[49], was introduced to GeuSap and Sapphire, before substitution into YC2.60. This mutation did not have any significant effects on the photophysical properties of the proteins, such as quantum yield, extinction coefficient and steady state anisotropy (Supplementary Table 6 and Supplementary Fig. 3). Sapphire-A206KΔ11 or GeuSap-A206KΔ11 genes were amplified by PCR, using a sense codon with NcoI site and an antisense codon with SphI site. We substituted GeuSap or Sapphire for eCFP of YC2.60 and constructed on pET28a at NcoI/XhoI site, with a polyhistidine tag on the C-terminus. We refer to these constructs as GeuSapVC2.60/pET28 and SapVC2.60/pET28, respectively. Constructs SapVC3.60/pET28 and GeuSapVC3.60/pET28 were made in a similar way. For the construction of the Cameleon-based sensors for mammalian expression, Calmodulin-M13-cp173Venus from the original YC2.60/3.60 plasmid was amplified by PCR with the sense primer containing an SphI site and the antisense primer containing a stop codon followed by an EcoRI site. SapA206KΔ11/GeuSapA206KΔ11 gene was amplified by PCR, with a sense codon containing HindIII site and an antisense codon containing SphI site. The amplified fragments of SapA206KΔ11/GeuSapA206KΔ11 and Calmodulin-M13-cp173Venus were then used in a third PCR reaction as both templates and primers of the spliced final gene. The spliced PCR product was digested with restriction enzymes HindIII and EcoRI, and subcloned into pcDNA3 (Invitrogen).

A complete list of primers used in this work can be found in Supplementary Table 7.

**Protein expression in *E. coli* and purification**. Recombinant proteins with polyhistidine tags were expressed in JM109(DE3) bacterial strain at 20°C, induced with 100 μM IPTG, then harvested and resuspended in TN buffer (50 mM Tris/HCl buffer pH7.5, 300 mM NaCl). cOmplete™ Mini EDTA-free Protease Inhibitor Cocktail (Sigma-Aldrich, St. Louis, MO) (1 tablet/10 ml), DNAse (25 U/ml, Sigma-Aldrich), MgCl$_2$ (1 mM) and lysozyme (1 mg/ml, Sigma-Aldrich) were also added. The resuspension was frozen overnight at −20 °C. The following day it was thawed at room temperature, centrifuged at 40,000 × g at 5 °C for 10 min to remove cell debris, and purified by nickel affinity chromatography (Qiagen, Hilden, Germany). For Ca$^{2+}$ sensors, proteins were further purified by size exclusion chromatography with Superdex 200 (GE Healthcare, Chicago, IL). Buffer composition for the size exclusion chromatography was 10 mM HEPES (pH 7.4), 150 mM NaCl and 100 μM EDTA. The buffer was exchanged with desalting column (PD-10, GE Healthcare) to HEPES buffer (10 mM, pH 7.4) supplemented with 150 mM NaCl, for all proteins unless otherwise noted.

**Steady state spectroscopy**. Absorption was measured with a spectrophotometer (Lambda 40 UV-Vis, PerkinElmer, Waltham, MA) and fluorescence with a fluorometer (Fluorolog 3, Horiba, Kyoto, Japan). Data were collected with Fluor-Essence v. 3.5. The molar extinction coefficient was experimentally determined as previously described[50]. Briefly, we determined the concentration of properly folded proteins by using 0.1 M NaOH as denaturant and the reference extinction coefficient value of 44,000 M$^{-1}$ cm$^{-1}$ at 447 nm. This concentration and the absorption at 400 nm were then used to calculate the molar extinction coefficient at 400 nm. We calculated the proper folding ratio (Supplementary Table 2), based on the ratio of concentration estimated with the alkaline method[50] over the concentration calculated based on the theoretical extinction coefficient at 280 nm. The theoretical extinction coefficient at 280 nm was calculated based on the amino acid sequence and assuming no S-S bonding[51].

Fluorescence quantum yield (Φ) for Sapphire mutants was calculated with the comparative method, using the quantum yield of wild type Sapphire (0.64) as a reference[27] at OD 0.1 at 400 nm. Steady state anisotropy was measured with a fluorometer (Fluorolog 3).

**Lifetime measurement in cuvette**. The fluorescence decay times of all constructs at the nanosecond time scale were determined by TCSPC technique. The frequency-doubled output (400 nm, 8.18 MHz, 2 ps FWHM) of a mode-locked Ti: Sapphire laser (Tsunami, Spectra Physics, Santa Clara, CA) was used as excitation source. The linearly polarized excitation light was rotated to a vertical direction by the use of a Berek compensator (New Focus) in combination with a polarization filter and directed onto the samples. The samples were placed in a quartz cuvette (10 mm path length), sealed by a Teflon stopper, and then mounted on the device. The emission was collected in lateral excitation configuration. The emission passed through a polarized plate with the tilt angle of 54.7° (magic angle), was introduced into a monochromator (Scientech 9030), and detected by a microchannel plate photomultiplier tube (MCP-PMT, R3809U-51, Hamamatsu). The fluorescence decay histogram in 4096 channels was obtained with a time correlated single photon timing PC module (SPC 830, Becker & Hickl). The decays were recorded with 10,000 counts in the peak channel, in time windows of 20 ns (GeuSap) and 40 ns (Sapphire). Data were collected with SPCM (v. 9.77, Becker & Hickl), and analysed with a time-resolved fluorescence analysis (TRFA v. 1.4) software[52], based on iterative reconvolution of the data with the instrumental response function (IRF). The full width at half- maximum (FWHM) of the IRF was typically in the order of 42 ps.

Assuming no other relaxation processes take place, rate constants of radiative ($k_r$) and nonradiative decay ($k_{nr}$) were calculated by using the equations:

$$\Phi = \frac{k_r}{k_r + k_{nr}} \tag{7}$$

$$\tau = \frac{1}{k_r + k_{nr}} \tag{8}$$

Solving for $k_r$ and $k_{nr}$, we get:

$$k_r = \frac{\Phi}{\tau} \tag{9}$$

$$k_{nr} = \frac{1}{\tau} - k_r \tag{10}$$

**Acceptor photobleaching experiment**. For the acceptor photobleaching, we imaged GeuSV11.5 and SV11.5 immobilized on Ni-NTA agarose beads (Qiagen) with a laser scanning microscope (Fluoview FV1000; Olympus, Tokyo, Japan), using an oil immersion objective lens (UPLSAPO 60x O, NA:1.35, Olympus). The acceptor was photobleached by using 488 nm laser at maximum power and the main dichroic mirror DM405/488. We, then, performed spectral imaging on the beads by exciting the donor with 405 nm laser and detecting in the range of 490–590 nm with 5 nm step size. Data were collected and exported with Fluoview software (v. 4.2c).

**Microscope setup**. We built a laser-scanning confocal microscope equipped with TCSPC units (Supplementary Fig. 9). A galvanometric mirror scanner (TILL Yanus IV digital scanner, FEI Munich, Gräfelfing, Germany,) was connected to the back port of the microscope body (IX70, Olympus). For excitation, a 405-nm pulsed diode laser (LDH-P-C-405, PicoQuant, Berlin, Germany) and 485-nm pulsed diode laser (LDH-D-C-485, PicoQuant) were installed on the microscope. Excitation beams were introduced into the scanner by reflecting on a dichroic mirror 405/488/561/640 (Chroma Technology GmbH, Olching, Germany). A water immersion objective (UPLSAPO 60XW, NA 1.2, Olympus) was used for all imaging experiments in this work. For detection, a H560LPXR dichroic mirror (AHF GmbH, Tübingen, Germany) splits the emitted fluorescence reflecting all incoming light below 560 nm. With the addition of a LP530 filter (AHF analysentechnik AG, Tübingen, Germany), the detection range is set as 530–560 nm. A polarizing beam splitter (CCM1-PBS251, Thorlabs, Newton, NJ) is placed after the filter to split emission to parallel and perpendicular. Emission was detected by two hybrid photomultiplier tubes (PMA Hybrid 40, PicoQuant). Decays were obtained by TCSPC (Hydraharp 400, PicoQuant). Data was acquired with SymPhoTime 64 (PicoQuant), imported with PAM software[53], and exported as TIFF images for further analysis. The PAM software is available as source code, requiring MATLAB or a precompiled, standalone distribution for Windows or MacOS at http://www.cup.uni-muenchen.de/pc/lamb/software/pam.html or hosted in Git repositories under http://www.gitlab.com/PAM-PIE/PAM and http://www.gitlab.com/PAM-PIE/PAMcompiled. Sample data are provided under http://www.gitlab.com/PAM-PIE/PAM-sampledata. A detailed manual is found at http://pam.readthedocs.io. Further analysis and fittings were performed with in-house-developed Python scripts.

**Time-resolved anisotropy**. Fluorescence decay analyses and time-resolved anisotropy experiments of GeuSV11.5 and GeuSapVC2.60 were performed with the home-built microscope. We used 405 or 485 nm lasers for excitation. The decays were recorded with 10$^4$ counts (for lifetime measurements) or 10$^5$ counts (for anisotropy) in 3125 channels, in time windows of 50 ns corresponding to 16 ps per channel. The IRF was measured with either SeTau-405-NHS (SETA BioMedicals, Urbana, IL), or Atto488 (ATTO-TEC GmbH, Siegen, Germany), diluted in saturated aqueous KI and excited with 405 or 485 nm lasers. Data analysis and fittings were performed with in-house-developed Python scripts.

Anisotropy was calculated based on the equation:

$$r_{corr}(t) = f_{corr} \cdot r_{obs}(t) = f_{corr} \cdot \frac{I_{\parallel}(t) - G \cdot I_{\perp}(t)}{I_{\parallel}(t) + 2 \cdot G \cdot I_{\perp}(t)} \tag{11}$$

where $r_{obs}(t)$ is measured anisotropy at time $t$, $r_{corr}(t)$ is anisotropy at time t after correction, $I_{\parallel}(t)$ is the parallel signal, $I_{\perp}(t)$ is the perpendicular signal, $G$ is a compensation factor for different polarization detection efficiencies, and $f_{corr}$ is a correction factor for the depolarization induced by high-NA objectives (NA > 0.3)[54,55]. $G$ factor was estimated by using fluorescent dye SeTau-405-NHS (SETA BioMedicals), which has similar absorption and emission maxima as Sapphire (405 nm and 518 nm, respectively), as a reference. Briefly, parallel and perpendicular signals of SeTau-405-NHS were measured with 60x objective and $G_{60x} = I_{\parallel}^{60x}/I_{\perp}^{60x}$ was calculated. Steady state anisotropy of SeTau-405 can be assumed to be 0, since it is a small fluorophore (548.97 Da) with long lifetime (9.3 ns in water). $f_{corr}$ was estimated by using a low-NA objective lens (UPLSAPO4X, NA 0.16, Olympus) and by measuring anisotropy of free Sapphire in solution as follows. First, $r_{corr}$ of Sapphire was acquired assuming $f_{corr,4x} = 1$ with the low-NA objective. Then, the same measurement was repeated with the 60x objective to get $r_{obs}$, and $f_{corr,60x}$ was calculated from the ratio of $r_{corr}$ over $r_{obs}$.

**Ca$^{2+}$ and pH titration**. Ca$^{2+}$ titration of GeuSapVC2.60 and GeuSapVC3.60 was performed in vitro on the microscope by using reciprocal dilutions of Ca$^{2+}$-free and Ca$^{2+}$-saturated buffers from Calcium Calibration Buffer Kit #1 (Thermo Scientific, Rockford, IL), according to manufacturer's instructions. Steady-state anisotropy was calculated as the average of ~25 images. The values of Hill coefficient, K'$_d$, minimum and maximum anisotropy were determined by fitting with Hill function using non-linear least squares with a home-written Python script. For pH titration of GeuSapVC2.60 we prepared the following series of buffers: 50 mM acetate (pH 5–5.5), 50 mM HEPES (pH 6–8), 50 mM Glycine (pH 8.5–10), supplemented with 150 mM NaCl.

For in situ Ca$^{2+}$ titration we performed imaging on live HeLa cells as previously described[56,57]. More specifically, we used Ca$^{2+}$-free and Ca$^{2+}$-saturated buffers from Calcium Calibration Buffer Kit #1 (Thermo Scientific) supplemented by 10 μM rotenone (Sigma Aldrich), 5 μM cyclopiazonic acid (Sigma–Aldrich), 1.8 mM 2-deoxy-D-glucose (Sigma Aldrich), and 10 μM 4-bromo-A23187 (Sigma Aldrich), a non-fluorescent calcium ionophore. The buffers contained increasing concentrations of CaEGTA (0–9 mM increasing by steps of 1 mM, 9.8 mM, and 10 mM). For the step of 0 mM CaEGTA the cells were incubated for 10 min before imaging, to allow equilibration of Ca$^{2+}$ across the membrane. For the following steps the incubation time was limited to 5 min, and to 2 min for the last step of 10 mM CaEGTA. Steady-state anisotropy calculation and fitting were performed the same way as for in vitro titration.

**Donor bleed-through compensation in angular displacement evaluation**. To calculate the angular displacement of the $Ca^{2+}$ sensor, donor bleed-through was compensated as follows.

Donor ($r_d$) and sensitized acceptor ($r_a$) anisotropies are described by:

$$r_d = \frac{I_{d\parallel} - I_{d\perp}}{I_{dt}} \rightarrow I_{d\parallel} - I_{d\perp} = r_d \cdot I_{dt} \tag{12}$$

$$r_a = \frac{I_{a\parallel} - I_{a\perp}}{I_{at}} \rightarrow I_{a\parallel} - I_{a\perp} = r_a \cdot I_{at} \tag{13}$$

where $I_{d\parallel}$ and $I_{d\perp}$ are parallel and perpendicular signals from donor in the acceptor channel, $I_{a\parallel}$ and $I_{a\perp}$ are parallel signals from sensitized acceptor, and $I_{dt}$ and $I_{at}$ are total signal from donor and sensitized acceptor in the acceptor channel.

Observed anisotropy is:

$$r_{obs} = \frac{(I_{d\parallel} + I_{a\parallel}) - (I_{d\perp} + I_{a\perp})}{I_{dt} + I_{at}} \rightarrow r_{obs} = \frac{(I_{d\parallel} - I_{d\perp}) + (I_{a\parallel} - I_{a\perp})}{I_{dt} + I_{at}} \tag{14}$$

Substituting eqs. 12 and 13 in Eq. 14:

$$r_{obs} = \frac{r_d \cdot I_{dt} + r_a \cdot I_{at}}{I_{dt} + I_{at}} \tag{15}$$

Donor fraction in the acceptor channel ($f_d$) is described by:

$$f_d = \frac{I_{dt}}{I_{dt} + I_{at}} \rightarrow I_{dt} + I_{at} = \frac{I_{dt}}{f_d} \rightarrow I_{at} = \frac{1 - f_d}{f_d} \cdot I_{dt} \tag{16}$$

Substituting Eq. 16 in Eq. 15 gives:

$$r_{obs} = \frac{r_d \cdot I_{dt} + r_a \cdot \left(\frac{1 - f_d}{f_d} \cdot I_{dt}\right)}{\frac{I_{dt}}{f_d}} \rightarrow r_{obs} = r_d \cdot f_d + r_a \cdot (1 - f_d) \rightarrow r_a = \frac{r_{obs} - r_d \cdot f_d}{1 - f_d} \tag{17}$$

**Cell culture and transfection**. HeLa cells were maintained in Dulbecco's modified Eagle medium (DMEM) without phenol red (Life Technologies, Carlsbad, CA), supplemented with 10% fetal bovine serum (Life Technologies) and 50 μg/ml gentamycin (Life Technologies) at 37 °C under humidified 5% $CO_2$ atmosphere. Cells were seeded on a glass bottom 29 mm dish (#1.5) (Cellvis, Mountain View, CA) two days before imaging, and transfected with either GeuSapVC2.60/pcDNA3 or GeuSapVC3.60/pcDNA3 according to the manufacturer's protocol by using X-tremeGENE 9 DNA Transfection Reagent (Sigma-Aldrich). Cells were imaged 48 hours after seeding and washed 3 times with HBSS (1.26 mM $CaCl_2$, 0.49 mM $MgCl_2$, without phenol red; Life Technologies) right before imaging.

**$Ca^{2+}$ imaging**. Imaging was performed with the home-built confocal microscope with 405 nm excitation. Cells were challenged with 100 μM histamine during image acquisition. Images were initially processed with ImageJ (https://imagej.nih.gov/ij/) and data analyzed with in-house-developed Python scripts.

For the generation of intensity-modulated display (IMD) anisotropy cell images, the anisotropy was computed from Eq. 11. For visualization purposes, the resulting frames were resized from 200 × 200 to 400 × 400 pixels, and convoluted with a Gaussian filter. The latter had a temporal and spatial standard deviation of 2 frames and 0.75 pixels, respectively. The anisotropy frames were then transformed to HSV format: The Hue was determined from the anisotropy (for visualization purposes, the values beyond the relevant range [0,0.12] were set to the nearest bounds, the Saturation was set to maximum, and the Value was controlled by the mean intensity frame.

**Reporting summary**. Further information on research design is available in the Nature Research Reporting Summary linked to this article.

## Data availability
The raw data acquired in this study are available at 10.6084/m9.figshare.14248487.v2. Source data are provided with this paper.

## Code availability
The software package PAM[53] is available as source code, requiring MATLAB to run, or as pre-compiled standalone distributions for Windows or MacOS at http://www.cup.uni-muenchen.de/pc/lamb/software/pam.html or hosted in Git repositories under http://www.gitlab.com/PAM-PIE/PAM and http://www.gitlab.com/PAM-PIE/PAMcompiled. Individual Python scripts are available upon request.

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

## Acknowledgements

The authors thank Dr. R.Y. Tsien and Dr. A. Miyawaki for providing plasmids, Dr. H. Hosoi for fruitful discussions, and Ms. E. Deridder for technical assistance. This work was partly supported by Interdisciplinaire Onderzoeksprogramma (IDO) KU Leuven (IDO/12/020) and the Category 1 research grant from KU Leuven (C14/16/053). DL acknowledges KU Leuven Facultaire Luik Onderzoeksfonds (FLOF) for financial support.

## Author contributions

D.L. and H.M. conceived and designed the experiments. D.L. performed the experiments and analyzed data. E.F. assisted spectroscopic analyses. G.S.F., Q.C., J. Hendrix and J. Hofkens designed and built-up the home-build microscope, assisted microscopic experiments and provided technical knowledge. D.L. and H.M. wrote the manuscript and all other authors provided input on the manuscript.

## Competing interests

The authors declare no competing interests.
