## [Peer Review File · Nature Communications]

Reviewers' Comments:

Reviewer #1:

Remarks to the Author:

This is an interesting methodological paper describing a relatively novel approach to monitor FRET changes in a Ca²⁺ sensor. As an example the Author described the behaviour of a novel Ca²⁺ indicator, first in a series of in vitro tests and then in live HeLa cells challenged with the agonist histamine.

Though the biophysical characterisation of the new sensors is quite accurate, the biological application and the potential advantages provided by this approach for the study of Ca²⁺ signalling are quite superficial. Key parameters such as signal to noise ration, pH sensitivity and in situ Ca²⁺ affinity have not been investigated and compared to classical genetically encoded Ca²⁺ probes. Given the present major interest of most groups for in vivo studies it would appear necessary to investigate the behaviour of this novel probes in two photon confocal microscopy. Without a deeper analysis of the advantage and disadvantages of this approach in living cells in my opinion this manuscript is of modest interest for the community of cell biologists or physiologists.

Reviewer #2:

Remarks to the Author:

The work submitted by Laskaratou et al presents a new approach (called FADED for FRET-induced Angular displacement Evaluation via Dim donor) for the measurement of genetically encoded FRET biosensors. The idea is to quantify the FRET-induced angular displacement between donor and acceptor by monitoring sensitized acceptor anisotropy. Usually this measurement is difficult to carry out because of donor bleed-through but here the very good idea is to use a dim fluorescent donor. The use of a dim donor seems strange because one suppose that the FRET is directly related to donor quantum yield but in fact its influence is weak since it is proportional to the one sixth power of quantum yield. Then in this paper they use a dim Sapphire mutant with around one fifth of the common Sapphire quantum yield preventing detectable spectral bleed through but with a very weak change in FRET properties. It is then easy to measure Venus sensitized acceptor anisotropy in the spectral band of Venus after having excited the dim fluorescent donor. It is a very nice approach since the measurement is directly related to donor-acceptor orientation making a new parameter to be investigated when one wants to develop a new biosensor.

The work presented is well executed with sufficient control despite the use of only one cellular situation monitoring calcium activation using histamine with a calmodulin developed biosensor using FADED method as a proof-of-concept. I'm sure that it will be of high interest in the microscopy community, particularly for FRET biosensors developers.

I have few concerns regarding this manuscript:

- The possibility to show FRET between dim donor and acceptor can be measured by the donor lifetime even if this lifetime is short (by using the spectral band of Sapphire excluding Venus fluorescence). This would be included in the manuscript to strongly support that FRET occurs.
- The use of homoFRET for FRET biosensors is missed in the discussion. This has been used by the group of Jin Zhang recently.
- I'm surprised by the different values of donor spectral bleed-through presented in figure 4f and g. It seems that the decrease of spectral bleed through in the calmodulin example is less than expected when using conventional tandems in fig 2. This is not really discussed. Can you comment that?

Reviewer #3:

Remarks to the Author:

In this manuscript (NCOMMS-20-37794-T), Laskaratou et al investigate the FRET-induced

depolarization of a FRET pair (GeuSap-venus) mutations with a dim (low quantum yield) donor (GeuSap; a mutant of sapphire) using steady-state and time-resolved anisotropy measurements. As a proof of principle, the authors investigate the sensitivity of a calmodulin-domain based FRET sensor for Ca²⁺ sensing. The main readout in these measurements is the steady-state anisotropy of the acceptor under the excitation of the dim donor as well as the relative donor-acceptor dipoles angle. Using these measurements, the authors were able to quantify the in vitro binding constant of Ca²⁺ and the corresponding Hill coefficient. Similar Ca²⁺ measurements were carried out in HeLa cells where the FRET construct was expressed in the cytosol. While there is still a spectral overlap of the donor and acceptor in these studies, the low fluorescence quantum yield of the donor indicate low level of bleed-through.

The authors did an excellent job concerning the control experiments in this well-written manuscript. The author refer to this concept of anisotropy-based FRET measurements as a "novel" approach. However, Currie et al (JPC B, 2017) and Leopold et al (JPC B, 2019) have reported on using time-resolved anisotropy of donor-acceptor pairs for FRET analysis as a measure of macromolecular crowding sensing.

While these constructs with dim donor may be suited for steady-state anisotropy imaging, they are unlikely to be useful for FLIM imaging due to the low quantum yield of the donor, which is the main observable in these type of measurements.

What is also not clear in these measurements on living cells is whether the authors have assessed the intrinsic autofluorescence of cellular coenzymes (e.g., NADH and flavins). This is especially important due the 400-nm excitation and the low fluorescence quantum yield of the donor.

Additional comments to the authors are outlined below:

(1) If I am not mistake, there is no mention of the corresponding FRET efficiency of these constructs anywhere in this manuscript. This seems a bit strange since FRET is in the title. This might be important to correlate between the observed anisotropy observables and the FRET efficiency in order to elucidate the underlying mechanism for Ca²⁺-sensing for example.

(2) What is the corresponding Forster distance of these FRET pairs investigated here? Please elaborate.

(3) Since the fluorescence decay of GeuSap (donor) is a biexponential, which component was used in Equation (3) for FRET analysis?

(4) In discussing Equation (3) for a FRET pair with a dim donor, the authors suggest that this concept would work if the D-A are close enough (but how close with respect to the conventional 10-nm maximum distance?) and if the FRET rate is equivalent to the fluorescence decay rate of the donor. How valid is that in these constructs and how is the estimated FRET time scale?

(5) What was the rationale for carrying out the acceptor-photobleaching experiments on immobilized GeuSV11.5 and SV11.5 on NiNTA beads? Please elaborate.

(6) For a steady-state anisotropy plateau of 0.14, which is much smaller dynamic range consider the theoretical limit (0.4). I am wondering if this would limit the potential applications of these FRET constructs. Please elaborate.

(7) Equation (4) predicts a single-exponential anisotropy decay. Yet, Figure 3(d) shows a multi-exponential anisotropy decay for GeuSV11.5 under 405-nm excitation. How do the authors explain/model that behavior?

(8) Does the spectral unmixing required here limit the application of these FRET constructs? Please

elaborate.

(9) The authors speculate that expressing these FRET pairs in living cells would influence their measured rotational time due to the high viscosity. Yet, no references were cited or cytosolic (where some of the measurements were made here) viscosity values was mentioned.

(10) When discussing the work of Thomas Jovin and his group, the authors should cite Warren et al (Int. J. Mol. Sci. 2015) when discussing anisotropy measurements for homo-FRET and hetero-FRET.

We are grateful to the editor for spending time on the manuscript and the referees for their helpful comments; our replies to the specific suggestions follow below. Criticism from reviewers is in black, our reply is in blue, and changes in the manuscript are highlighted in red.

Kind regards,

The authors

Reviewer #1 (Remarks to the Author):

This is an interesting methodological paper describing a relatively novel approach to monitor FRET changes in a Ca²⁺ sensor. As an example the Author described the behaviour of a novel Ca²⁺ indicator, first in a series of *in vitro* tests and then in live HeLa cells challenged with the agonist histamine.

Though the biophysical characterisation of the new sensors is quite accurate, the biological application and the potential advantages provided by this approach for the study of Ca²⁺ signalling are quite superficial. Key parameters such as signal to noise ration, pH sensitivity and *in situ* Ca²⁺ affinity have not been investigated and compared to classical genetically encoded Ca²⁺ probes.

Thank you for pointing out these additional sensor parameters, they are indeed important for the complete characterization of this new type of sensors for biological applications. We included signal-to-noise ratio and pH titration (see Supplementary Figures) as per your suggestion. The pH titration showed that the sensor responds stably at pH 7 and above, in agreement with the results reported on classical genetically encoded Ca²⁺ probes (see for example Nagai et al., PNAS 2004).

A typical method for Ca²⁺ calibration is the use of fluorescent dyes, such as Fura2, to generate a curve and correlate this to emission-based data measured *in situ*. Since this is not applicable to our anisotropy-based measurements, we attempted *in situ* Ca²⁺ affinity measurements by following the method proposed by Rong et al. (Sci. Rep. 2017). This method was detrimental to the cell condition, as can be seen from images below. For this reason, we do not wish to include data from this experiment in the manuscript, as it is clear to us that it does not reflect cellular conditions any close to normal. We estimated the Ca²⁺ concentration in live cells based on the *in vitro* calibration in this work, and the values were consistent with previous reports on classical genetically encoded Ca²⁺ probes (Nagai et al., PNAS 2004, Miyawaki et al., PNAS 1999), so we believe this method is reliable enough.

More specifically about the signal-to-noise ratio, we calculated the signal-to-noise ratio based on the general definition of mean fluorescent signal for parallel and perpendicular channels and their corresponding standard deviations, and included the results in the supplementary material.

We think that comparison with classical ratiometric GECIs in terms of signal-to-noise ratio is both technically difficult and will shift the reader's attention from the main message of the paper. The technical difficulty lies on the fact that anisotropy can take negative or zero values. A metric often used in the field to compare between Ca²⁺ biosensors is $\Delta R/R_0 = (R - R_0)/R_0$ (referred to as signal-to-background ratio, see for example Koldenkova et al., Mol Cell Res 2013), where $R - R_0$ is the ratio change upon Ca²⁺ change and R_0 is the average baseline ratio under resting conditions. Another measure to compare is dynamic range, defined as R_{max}/R_{min} for ratiometric indicators, where R_{max} and R_{min} are the maximal and minimal FRET ratios under conditions of Ca²⁺ saturation or depletion, respectively (Koldenkova et al., Mol Cell Res 2013). We do not believe these metrics can be applied to anisotropy, because anisotropy can take values of zero or negative, which have physical

meaning, but render these ratio calculations meaningless. Strictly speaking about SNR, it depends largely on measuring conditions, thus it is not possible to compare with reported values, and also not possible to normalize conditions for different types of FRET pairs side-by-side. More importantly, we have developed the Ca²⁺ sensors in this work as a proof of principle for FADED, and certainly not as improved versions of classical genetically encoded Ca²⁺ probes.

Regarding the reviewer's doubts about the potential advantages of this approach to study Ca²⁺, we would like to point out that anisotropy is an absolute quantity that does not depend on measuring parameters. This allows for easy correlation between *in vitro* calibration and imaging in live cells, as we explain in the discussion. Of course, we developed these Ca²⁺ sensors as a means to promote the main message of the paper, which is the new mode of FRET sensing based on angular displacement, and clearly saw that the concept is working also in live cells. More generally put, this kind of probe senses changes in relative orientation between donor and acceptor, as opposed to classical FRET biosensors, which provide information on the donor-acceptor distance. In that sense, this mode of sensing provides complementary information to distance, and as such is not advantageous or disadvantageous *per se*. However, in some cases where the change in orientation outweighs change in distance (Campbell, Anal. Chem. 2009), we speculate that this type of sensors can prove beneficial over conventional distance-based biosensors and provide insightful biological information. We are planning to investigate those cases in the future to show the advantageous potential of FADED.

Given the present major interest of most groups for *in vivo* studies it would appear necessary to investigate the behaviour of this novel probes in two photon confocal microscopy. Without a deeper analysis of the advantage and disadvantages of this approach in living cells in my opinion this manuscript is of modest interest for the community of cell biologists or physiologists.

Thank you for the interesting suggestion, we agree that applications in living organisms are important. Two-photon microscopy would indeed be appealing for *in vivo* applications of this idea. However, this work is the very first introduction to the concept and provides a proof of principle. In addition, installing a femtosecond laser to our microscope setup is a big investment. Therefore, we reserve two-photon microscopy and *in vivo* applications for the future.

Reviewer #2 (Remarks to the Author):

The work submitted by Laskaratou et al presents a new approach (called FADED for FRET-induced Angular displacement Evaluation via Dim donor) for the measurement of genetically encoded FRET biosensors. The idea is to quantify the FRET-induced angular displacement between donor and acceptor by monitoring sensitized acceptor anisotropy. Usually this measurement is difficult to carry out because of donor bleed-through but here the very good idea is to use a dim fluorescent donor. The use of a dim donor seems strange because one suppose that the FRET is directly related to donor quantum yield but in fact its influence is weak since it is proportional to the one sixth power of quantum yield. Then in this paper they use a dim Sapphire mutant with around one fifth of the common Sapphire quantum yield preventing detectable spectral bleed through but with a very weak change in FRET properties. It is then easy to measure Venus sensitized acceptor anisotropy in the spectral band of Venus after having excited the dim fluorescent donor. It is a very nice approach since the measurement is directly related to donor-acceptor orientation making a new parameter to be investigated when one wants to develop a new biosensor.

The work presented is well executed with sufficient control despite the use of only one cellular situation monitoring calcium activation using histamine with a calmodulin developed biosensor using FADED method as a proof-of-concept. I'm sure that it will be of high interest in the microscopy community, particularly for FRET biosensors developers.

I have few concerns regarding this manuscript:

- The possibility to show FRET between dim donor and acceptor can be measured by the donor lifetime even if this lifetime is short (by using the spectral band of Sapphire excluding Venus fluorescence). This would be included in the manuscript to strongly support that FRET occurs.

We are grateful to the reviewer for this suggestion. We included an extra panel in Fig. 2 and calculations based on this new data.

- The use of homoFRET for FRET biosensors is missed in the discussion. This has been used by the group of Jin Zhang recently.

Thank you for bringing this work to our attention, we included it in the discussion.

- I'm surprised by the different values of donor spectral bleed-through presented in figure 4f and g. It seems that the decrease of spectral bleed through in the calmodulin example is less than expected when using conventional tandems in fig 2. This is not really discussed. Can you comment that?

This is an interesting observation. This apparent discrepancy can be explained by the different FRET efficiencies between the tandem and the Ca^{2+} sensor. In fact, the calculated FRET efficiency for the tandem is 76%, while for the Ca^{2+} sensor this is only 40% (free form), or 52% at best (bound form). We included an explanation in the discussion.

Reviewer #3 (Remarks to the Author):

In this manuscript (NCOMMS-20-37794-T), Laskaratou et al investigate the FRET-induced depolarization of a FRET pair (GeuSap-venus) mutations with a dim (low quantum yield) donor (GeuSap; a mutant of sapphire) using steady-state and time-resolved anisotropy measurements. As a proof of principle, the authors investigate the sensitivity of a calmodulin-domain based FRET sensor for Ca^{2+} sensing. The main readout in these measurements is the steady-state anisotropy of the acceptor under the excitation of the dim donor as well as the relative donor-acceptor dipoles angle. Using these measurements, the authors were able to quantify the in vitro binding constant of Ca^{2+} and the corresponding Hill coefficient. Similar Ca^{2+} measurements were carried out in HeLa cells where the FRET construct was expressed in the cytosol. While there is still a spectral overlap of the donor and acceptor in these studies, the low fluorescence quantum yield of the donor indicate low level of bleed-through.

The authors did an excellent job concerning the control experiments in this well-written manuscript. The author refer to this concept of anisotropy-based FRET measurements as a "novel" approach. However, Currie et al (JPC B, 2017) and Leopold et al (JPC B, 2019) have reported on using time-resolved anisotropy of donor-acceptor pairs for FRET analysis as a measure of macromolecular crowding sensing.

Thank you for your comments and for bringing these two papers to our attention. We included them in the discussion. However, we would like to stress that the novelty of our approach lies in the use of a dim fluorescent donor and the almost-exclusive detection of acceptor signal. These factors make the quantification of angular displacement possible. This was simply not the focus of Currie et al (JPC B, 2017), nor Leopold et al (JPC B, 2019), who developed macromolecular crowding biosensors based on mCerulean-mCitrine FRET pair, and carried out time-resolved anisotropy measurements on those. Currie et al. did calculate the angle between absorption and emission dipole moments for these constructs, but did not account for donor bleedthrough in the detection band of 510-550 nm.

While these constructs with dim donor may be suited for steady-state anisotropy imaging, they are unlikely to be useful for FLIM imaging due to the low quantum yield of the donor, which is the main observable in these type of measurements.

This may be true, but in any case we did not intend to use these sensors for FLIM.

What is also not clear in these measurements on living cells is whether the authors have assessed the intrinsic autofluorescence of cellular coenzymes (e.g., NADH and flavins). This is especially important due the 400-nm excitation and the low fluorescence quantum yield of the donor.

Thank you for pointing out this important issue. We checked for autofluorescence influence as per your suggestion and included supporting data in the Supplementary section. We confirmed that autofluorescence signals are negligible compared to fluorescence originating from the sensor.

Additional comments to the authors are outlined below:

(1) If I am not mistake, there is no mention of the corresponding FRET efficiency of these constructs anywhere in this manuscript. This seems a bit strange since FRET is in the title. This might be important to correlate between the observed anisotropy observables and the FRET efficiency in order to elucidate the underlying mechanism for Ca²⁺-sensing for example.

Thank you for raising this issue, we have included the calculations of corresponding FRET efficiencies in the main text.

(2) What is the corresponding Förster distance of these FRET pairs investigated here? Please elaborate.

Calculating the Förster distance requires knowledge of the orientation factor κ^2 . This is assumed to be 2/3 often in the literature, meaning that the fluorophores can take all possible orientations. We do not believe this is the case with our constructs; in fact, the angular displacement calculated throughout this paper indicates this, and is also inextricably linked to the orientation factor. For these reasons, we do not wish to include Förster distance calculations in the manuscript, but in any case we include them here with the assumption $\kappa^2=2/3$. For GeuSap-cp173Venus, we calculated $R_0=4.4$ nm. For comparison, for the CFP-YFP pair $R_0=5$ nm.

(3) Since the fluorescence decay of GeuSap (donor) is a biexponential, which component was used in Equation (3) for FRET analysis?

We used the amplitude-weighted average lifetime, $\langle \tau \rangle = \sum_i a_i \tau_i$, where $\sum_i a_i = 1$. We also made it clear in the main text. Thank you for pointing this out.

(4) In discussing Equation (3) for a FRET pair with a dim donor, the authors suggest that this concept would work if the D-A are close enough (but how close with respect to the conventional 10-nm maximum distance?) and if the FRET rate is equivalent to the fluorescence decay rate of the donor. How valid is that in these constructs and how is the estimated FRET time scale?

In fact, this concept is valid regardless of D-A distance as long as there is detectable FRET signal, which is the case for all constructs throughout this work. More concretely, for the tandem construct GeuSV11.5 we calculated a FRET efficiency of 76%, meaning that D-A distance is much less than 10 nm (or even less than 4.4 nm calculated above, if we assume $\kappa^2=2/3$). We also calculated the FRET time constant as $\tau_T=0.29$ ns, which is shorter than the donor lifetime, $\tau_D^0=0.92$ ns. We carried out this analysis for the Ca²⁺ sensors as well, and the FRET time constant was slightly longer than donor lifetime for the free form, but equivalent to donor lifetime for the bound form (1.38 ns and 0.85 ns, respectively). Based on the above, we consider this statement to be valid. We have included these analyses in the main text.

(5) What was the rationale for carrying out the acceptor-photobleaching experiments on immobilized GeuSV11.5 and SV11.5 on NiNTA beads? Please elaborate.

The rationale behind the acceptor photobleaching experiments was to provide first evidence for FRET. Acceptor photobleaching is a standard experiment for quickly verifying if FRET takes place, and is based on the principle that in the absence of the acceptor FRET does not happen. The decreasing sensitized acceptor signal

due to elimination of the acceptor species by photobleaching is accompanied by a reciprocal increase in the direct emission signal from the donor. For more information, please check Karpova et al., J. Microsc. 2003.

(6) For a steady-state anisotropy plateau of 0.14, which is much smaller dynamic range consider the theoretical limit (0.4). I am wondering if this would limit the potential applications of these FRET constructs. Please elaborate.

In this type of sensors, the change in anisotropy is strongly system dependent, and is sensitive to the angular displacement between donor and acceptor, among other system parameters. The theoretical maximum range is 0.6, assuming a full anisotropy range of [-0.2,0.4]. In reality, we cannot make full use of this range, because it corresponds to the extreme cases of fluorophores in tandem (0.4) and perpendicular (-0.2). For our Ca^{2+} sensor in particular, the anisotropy range *in vitro* is found to be [-0.03,0.10], and is sufficient for sensing purposes in live cells (see Fig. 5 of main text). Other systems may turn out to have larger or smaller range than this, depending on their architecture.

(7) Equation (4) predicts a single-exponential anisotropy decay. Yet, Figure 3(d) shows a multi-exponential anisotropy decay for GeuSV11.5 under 405-nm excitation. How do the authors explain/model that behavior?

This is indeed an interesting point. As mentioned in the main text, in the case of donor excitation a very fast component attributable to FRET is visible. This is followed by a slow component, which decays with a very similar time constant as direct acceptor excitation (see parallel decays after e.g. 5 ns in Fig. 3d). The fast component was not fitted, since the time range was beyond the resolution of our system. For the above reasons, we only applied fitting with a monoexponential model after 5 ns, taking into account only the slow component of this multi-exponential decay.

(8) Does the spectral unmixing required here limit the application of these FRET constructs? Please elaborate.

The spectral unmixing was used in this work only to evaluate the extent of donor bleedthrough. It is not required for anisotropy FRET measurements.

(9) The authors speculate that expressing these FRET pairs in living cells would influence their measured rotational time due to the high viscosity. Yet, no references were cited or cytosolic (where some of the measurements were made here) viscosity values was mentioned.

Thank you for mentioning this, we have cited several references which give cytosolic viscosity values and corresponding rotational diffusion times of FPs in viscous solution.

(10) When discussing the work of Thomas Jovin and his group, the authors should cite Warren et al (Int. J. Mol. Sci. 2015) when discussing anisotropy measurements for homo-FRET and hetero-FRET.

We have included this work in the discussion. Thank you for bringing it to our attention.

Reviewers' Comments:

Reviewer #1:

Remarks to the Author:

The authors have only partially answered to my concerns.

1. Concerning pH sensitivity they reported only the effect of pH changes in vitro, while it would have been more interesting if they did it on intact cells. There are plenty of ways to modify cytosolic pH in the physiological range.

2. Less convincing is the answer to my second question, the calibration in situ. I am perfectly aware that in situ calibration, particularly for cytosolic probes, are far from simple and it requires some time to establish the proper protocol. Nonetheless the results presented are quite strange and even more strange is the decision of the Authors to discard the approach because of some morphological changes of the cells upon addition of ionomycin. This ionophore is routinely used by many laboratories in the world and in the absence of a high level of extracellular free Ca^{2+} it is notoriously without any significant toxicity effect on cells (at least in the short term). The changes observed (particularly given that there is no information on the time of incubation) are not so dramatic. Add to this that a calibration with ionomycin is the less reliable approach and several other methods (e.g. from the use of a patch pipette, to selective permeabilization to ions of the plasma membrane) have been used by several authors. In more general terms, I agree with the statement of the authors that their paper is primarily a proof of principle (and in this respect I think they did an excellent job). However, simple minded cell biologist (the majority, I suspect, of the Nature Comm. readers) wish to know whether a different approach/method has advantages (or disadvantages) with respect to more traditional ones. I think that the Authors should try to make an effort to convince biologists about the advantages of the method proposed. Some of the sentences of their reply letter could help in this direction and could be included in the manuscript text.

3. Finally, a sentence concerning the potential applicability (or not) of the present approach to two photon microscopy would be, in my opinion, appropriate.

Reviewer #2:

Remarks to the Author:

The authors adequately answer to my queries.

Reviewer #3:

Remarks to the Author:

The authors did an excellent job addressing all of my concerns and therefore I am delighted to recommend this manuscript for publication in Nature Communications.

We are grateful to the editor for spending time on the manuscript and the referees for their helpful comments; our replies to the specific suggestions follow below. Criticism from reviewers is in black, our reply is in blue, and changes in the manuscript are highlighted in red.

Kind regards,

The authors

Reviewer #1 (Remarks to the Author):

The authors have only partially answered to my concerns.

1. Concerning pH sensitivity they reported only the effect of pH changes *in vitro*, while it would have been more interesting if they did it on intact cells. There are plenty of ways to modify cytosolic pH in the physiological range.

In general, we agree with the reviewer that experiments should be performed in the same context and environment as the sensor is used in (i.e. intact cells). However, in this specific case for pH sensitivity, we do not fully grasp why an *in situ* experiment would be beneficial. The purpose of pH sensitivity experiment is to show the stability of the sensor in certain pH range, and therefore we believe that the method *in vitro* is more appropriate, since we can precisely control pH and there are no secondary effects common in complex environments such as a cell. The *in vitro* titration has been performed for most of genetically encoded Ca²⁺ sensors, including YC2.6, one of the best classical Ca²⁺ sensors which has the same backbone as our sensor, and therefore we could directly compare our sensor to others.

There are indeed methods to modify pH *in situ*. However, if the anisotropy values *in situ* differ from those reported in the *in vitro* experiment, this should be attributed to some cellular phenomena acting synergistically, and not exclusively to the change in pH (see for example Speake et al., J Physiol 1998, where they increased the cytosolic pH in rat pancreatic cells from 7.2 to 7.7, and saw an increase in [Ca²⁺] as well). Based on this, the input to the sensor will not be free from cross-talk events in the cellular environment, therefore we also expect the output to be equally complicated. The interpretation of such complex output (albeit interesting in itself) is, however, out of the scope of this paper.

2. Less convincing is the answer to my second question, the calibration *in situ*. I am perfectly aware that *in situ* calibration, particularly for cytosolic probes, are far from simple and it requires some time to establish the proper protocol. Nonetheless the results presented are quite strange and even more strange is the decision of the Authors to discard the approach because of some morphological changes of the cells upon addition of ionomycin. This ionophore is routinely used by many laboratories in the world and in the absence of a high level of extracellular free Ca²⁺ it is notoriously without any

significant toxicity effect on cells (at least in the short term). The changes observed (particularly given that there is no information on the time of incubation) are not so dramatic. Add to this that a calibration with ionomycin is the less reliable approach and several other methods (e.g. from the use of a patch pipette, to selective permeabilization to ions of the plasma membrane) have been used by several authors.

We carefully considered the reviewer's comments about ionomycin and implemented a different protocol, which uses A23187/calcimycin (Thomas et al. Cell Calcium 2000, Ljubojević et al. Biophys. J. 2011) and has a much less dramatic effect on cell condition. We included the *in situ* titration results in the Supplementary section and added a comment in the Results section in the main text.

In more general terms, I agree with the statement of the authors that their paper is primarily a proof of principle (and in this respect I think they did an excellent job). However, simple minded cell biologist (the majority, I suspect, of the Nature Comm. readers) wish to know whether a different approach/method has advantages (or disadvantages) with respect to more traditional ones. I think that the Authors should try to make an effort to convince biologists about the advantages of the method proposed. Some of the sentences of their reply letter could help in this direction and could be included in the manuscript text.

We thank the reviewer for this suggestion. We have included an extra paragraph in the discussion, incorporating those sentences.

3. Finally, a sentence concerning the potential applicability (or not) of the present approach to two photon microscopy would be, in my opinion, appropriate.

We have included a sentence on the potential applicability to two-photon microscopy. Since we did not perform any 2P experiments, however, we hesitate to elaborate more.

Reviewers' Comments:

Reviewer #1:

Remarks to the Author:

The answer to my last questions are acceptable and accordingly I think the paper can be accepted as it is.